# Novel Non-Viral Vectors Based on Pluronic^®^ F68PEI with Application in Oncology Field

**DOI:** 10.3390/polym14235315

**Published:** 2022-12-05

**Authors:** Inês Silva, Cátia Domingues, Ivana Jarak, Rui A. Carvalho, Rosemeyre A. Cordeiro, Marília Dourado, Francisco Veiga, Henrique Faneca, Ana Figueiras

**Affiliations:** 1Laboratory of Drug Development and Technologies, Faculty of Pharmacy, University of Coimbra, 3000-548 Coimbra, Portugal; 2REQUIMTE/LAQV, Group of Pharmaceutical Technology, University of Coimbra, 3000-548 Coimbra, Portugal; 3Institute for Clinical and Biomedical Research (iCBR) Area of Environment Genetics and Oncobiology (CIMAGO), Faculty of Medicine, University of Coimbra, 3000-548 Coimbra, Portugal; 4Department of Life Sciences, Faculty of Sciences and Technology, University of Coimbra, 3000-548 Coimbra, Portugal; 5CNC—Center for Neuroscience and Cell Biology, University of Coimbra, 3004-517 Coimbra, Portugal; 6Institute for Interdisciplinary Research (IIIUC), University of Coimbra, 3000-548 Coimbra, Portugal; 7Center for Health Studies and Research of the University of Coimbra (CEISUC), Faculty of Medicine, University of Coimbra, 3000-548 Coimbra, Portugal; 8Center for Studies and Development of Continuous and Palliative Care (CEDCCP), Faculty of Medicine, University of Coimbra, 3000-548 Coimbra, Portugal

**Keywords:** polyethylenimine, Pluronics^®^, pDNA, non-viral vectors, cancer therapy

## Abstract

Copolymers composed of low-molecular-weight polyethylenimine (PEI) and amphiphilic Pluronics^®^ are safe and efficient non-viral vectors for pDNA transfection. A variety of Pluronic^®^ properties provides a base for tailoring transfection efficacy in combination with the unique biological activity of this polymer group. In this study, we describe the preparation of new copolymers based on hydrophilic Pluronic^®^ F68 and PEI (F68PEI). F68PEI polyplexes obtained by doping with free F68 (1:2 and 1:5 *w/w*) allowed for fine-tuning of physicochemical properties and transfection activity, demonstrating improved in vitro transfection of the human bone osteosarcoma epithelial (U2OS) and oral squamous cell carcinoma (SCC-9) cells when compared to the parent formulation, F68PEI. Although all tested systems condensed pDNA at varying polymer/DNA charge ratios (N/P, 5/1–100/1), the addition of free F68 (1:5 *w/w*) resulted in the formation of smaller polyplexes (<200 nm). Analysis of polyplex properties by transmission electron microscopy and dynamic light scattering revealed varied polyplex morphology. Transfection potential was also found to be cell-dependent and significantly higher in SCC-9 cells compared to the control bPEI25k cells, as especially evident at higher N/P ratios (>25). The observed selectivity towards transfection of SSC-9 cells might represent a base for further optimization of a cell-specific transfection vehicle.

## 1. Introduction

Cancer is the most prevalent mortality cause with growing incidence. The design of new, more selective and specific chemotherapeutics combined with advances in early diagnostics and the development of new, revolutionary therapies offer renewed hope for cancer patients. Growing insight into cancer biology had led to the development and implementation of gene therapy in anticancer therapy. The broad choice of therapeutic targets and the use of existing cellular machinery represent some of the advantages of therapeutic nucleic acid (NA)-like DNA over classical chemotherapy [1]. However, the efficacy of gene therapy is compromised by serious difficulties associated with therapeutic DNA in overcoming biological barriers, including low stability, impaired cell internalization, or immunogenicity. As a consequence, the search for an efficient, biocompatible vector for DNA delivery has attracted significant attention. Among many nanosystems currently in preclinical development, nanovehicles based on amphiphilic polymers such as Pluronics^®^ are particularly interesting options for the preparation of highly stable vehicles for delivery of NAs. Pluronics^®^ are composed of an inner hydrophobic poly(propylene oxide) (PPO) chain flanked by hydrophilic poly(ethylene oxide) chains [2]. Different combinations of these two components result in a wide range of physicochemical and biological properties, which can be explored for improved drug and gene delivery [2,3], enabling them to self-aggregate into stable micelles stabilized by non-covalent interactions within the hydrophobic core and hydrophilic shell, which provides protection against adverse interactions with biological compartments. From the drug/NA-delivery standpoint, the most interesting Pluronics^®^ are those that aggregate at low concentrations, which increases their stability against dissociation upon systemic dilution. Conjugation with a cationic polymer such as the much-studied polyethyleneimine (PEI) enables complexation and condensation of pDNA via electrostatic interactions. Excess polycationic chains with high polymer/DNA enables interactions with cell membranes and facilitate cell uptake and transfection. Unique biological properties of some Pluronics^®^ can be used to cross biological barriers or enhance cell internalization [4]. Additionally, they can inhibit drug efflux transporters and impair the mitochondrial activity of multidrug-resistant cancer cells, which can be exploited in combination therapy [5,6]. Because biological activity is the most pronounced in Pluronics^®^ with intermediate hydrophilic–lipophilic balance (HBL), the most attention on pDNA delivery has been focused on polyplexes based on Pluronics^®^ with HBL ~ 15 and/or low cmc that can form micelles under experimental conditions. Polyplexes formed by Pluronics^®^ such as P85 (HBL 16, cmc 0.04 %wt), P123 (HLB 8, cmc 0.025 %wt) or F127 (HLB 22, cmc 0.0035 %wt) conjugated with PEI have been extensively studied [7,8,9]. However, polyplexes of hydrophilic Pluronics^®^ with relatively high cmc such as F68 (HLB 29, cmc 0.4 %wt) grafted with PEI also exhibit noteworthy transfection activity that deserves additional attention [10,11,12]. Although biological effects of F68 in cancer cells have not been investigated in detail, owing to its hydrophilicity and biocompatibility, it is widely used as surfactant, as well as a solubilizing and emulsifying agent, in pharmaceutical and cosmetic industries. It can improve the oral availability of hydrophobic drugs [13] and stabilize liquid formulations containing biologicals such as antibodies and proteins, although the exact mechanism of the observed stabilizing activity remains unknown [14]. Recently, several antibody formulations containing F68 were approved for clinical application [15]. 

To fill gaps in the literature, in this study, we prepared a new copolymer by conjugation of F68 to branched PEI1.8k and studied their transfection capacity under sub-cmc conditions. Furthermore, the influence of the addition of free F68 to F68PEI was investigated. New F68PEI polyplexes with pCMV.Luc plasmid were physicochemically and morphologically characterized in detail of, and their capability to bind pDNA was determined. Finally, the transfection capacity of human bone osteosarcoma epithelial (U2OS) and oral squamous cell carcinoma (SCC-9) cells was evaluated.

## 2. Materials and Methods

### 2.1. Materials

Commercial Pluronic^^®^^ F68 (Mw 8400 Da), *N*,*N*′-dicarbonyldiimidazole (CDI), ethanol, sodium carbonate, resazurin, hydrocortisone and McCoy’s 5A cell medium were acquired from Sigma-Aldrich (St. Louis, MO, USA). Acetonitrile was purchased from Carlo Erba (Val de Reuil, France). Commercial branched PEI 99% (Mw 1800 Da) and acryloyl chloride were acquired from Alfa Aesar (Haverhill, MA, USA). Regenerated cellulose dialysis tubing (MWCO 3500 Da) was acquired from Orange Scientific (Braine l’Alleud, Belgium). Cellulose acetate syringe filters were purchased from Filter-Lab^^®^^ (Barcelona, Spain). A DNA plasmid kit was obtained from Qiagen (Hilden, Germany), and U2OS and SCC-9 cell lines were purchased from ATCC. DMEM:F12, fetal bovine serum (FBS) (10%), penicillin and streptomycin were obtained from Biowest (Nuaille, France). A DC protein assay was acquired from Biorad (Hercules, CA, USA). Resazurin sodium salt was obtained from Sigma-Aldrich (St. Louis, MO, USA).

### 2.2. Synthesis and Characterization of Copolymer F68PEI

Copolymer composed of Pluronic^®^ F68 and branched PEI (bPEI1.8k) was prepared by am adapted two-step synthetic protocol based on the activation of Pluronics^®^ by 1,1-carbonyldiimidazole [5]. Briefly, a solution of Pluronic^®^ F68 (2 g, 0.24 mmol) in dry acetonitrile (15 mL) was added dropwise to a solution of *N*,*N*′-dicarbonyldiimidazole (CDI) (0.4 g, 2.38 mmol) in dry acetonitrile (10 mL). The reaction mixture was heated at 40 °C for 4 h and stirred overnight at room temperature. Distilled water (15 mL) was added to the reaction mixture, which was subsequently dialyzed (3500 Da) against 50% ethanol (3 h) and 10% ethanol (3 h). After dialysis, acetonitrile was removed under reduced pressure, and the remainder was lyophilized. Activated F68CDI was obtained as a white solid (0.562 g). ^1^H NMR (δ-ppm; D_2_0): 1.19 (m, PPO, CH_3_), 3.4–3.7 (m, PPO CH, CH_2_), 3.74 (m, PEO CH_2_), 7.14 (m, CH Im), 7.67 (m, CH Im), 8.4 (m, CH Im).

In the next step, a solution of PEI (0.128 g, 0.07 mM in 10 mL of buffer) was added dropwise to a solution of F68CDI (0.56 g) in sodium carbonate buffer (0.1 M pH = 8.2, 10 mL). The resulting solution was mixed for 72 h at room temperature. The mixture was dialyzed overnight against water. The product, F68-PEI, was isolated as a white solid (0.5 g). ^1^H-NMR (δ-ppm; D_2_0): 1.19 (m, PPO, CH_3_), 2.6 (m, CH_2_CH_2_ PEI), 3.4–3.7 (m, PPO CH, CH_2_), 3.74 (m, PEO CH_2_).

The synthesized copolymers were physicochemically characterized by nuclear magnetic resonance (NMR) and Fourier transform–infrared spectroscopy (FT-IR). The proton nuclear magnetic resonance (^1^H-NMR) spectra of polymers were acquired using a Varian spectrophotometer operating at 500 Hz with a 3 mm indirect detection probe. Acquisition parameters comprised a 7.2 KHz sweep width, a 30° radiofrequency pulse, 3 s acquisition time and an interpulse delay of 10 s. Samples were prepared in 250 μL of D_2_O, and 32 scans were recorded for each sample. The spectra were analyzed with TopSpin (version 4.0.8, Bruker Biospin GmbH, Rheinstetten, Germany). For FT-IR analysis, spectra were recorded using a Jasco FT/IR-420 spectrometer. Lyophilized samples were used to acquire the spectra, and 32 scans were performed at a resolution of 2 cm^−1^ in the range of 4000–650 cm^−1^.

### 2.3. TNBS Assay

The amount of PEI in prepared F68PEI was evaluated by determining primary amine content using a trinitrobenzenesulfonic acid (TNBS) assay [16]. Standard solutions of bPEI1.8k (1–50 μg/mL) and the F68PEI test solution of were diluted with 0.1 M borate buffer (pH 9.5) to a total volume of 100 μL and pipetted into a 96-well plate. Then, 25 μL of TNBS (1 M aqueous solution diluted to 0.1% with borate buffer) was added to each well. Borate buffer was used as a blank sample. After 30 min, absorption was measured at 405 nm using a microplate reader (Synergy HT luminometer, Biotek, Winooski, VT, USA).

### 2.4. Buffering Capacity of F68PEI

The buffering capacity of F68PEI was evaluated by acid–base titration. The pH of the polymer solution (equivalent to 0.4 mg/mL of bPEI1.8k) was adjusted to 3 by the addition of HCl, and the solution was titrated with 0.1 M NaOH until reaching pH = 10. The pH values were recorded using a pH meter.

### 2.5. Preparation of Polyplexes

Polyplexes were prepared by the addition of pDNA (pCMV.Luc plasmid, 2.85 μg/mL) to the polymer solutions in ddH_2_O (pH 7.4). The resulting mixture was gently shaken for 15 s and incubated at room temperature for 15 min. Polyplexes were prepared at varying [polymer nitrogen]/[pDNA phosphate] (N/P) ratios ranging from 5 to 100. The concentration of F68PEI polymer in F68PEI-pDNA polyplexes ranged from 0.017 to 0.34 mg/mL. For the mixed polymer systems composed of F68PEI with the addition of free F68, the total polymer concentration was in the range of 0.05–1.01 mg/mL (F68PEI/F68 1:2, *w/w*) and 0.1–2.02 mg/mL (F68PEI/F68 1:5, *w/w*). Freshly prepared polyplexes were used for characterization and cell viability/transfection experiments.

### 2.6. Physicochemical Characterization of F68PEI Nanosystems

F68PEI nanosystems were extensively physicochemically characterized before and after pDNA complexation.

The mean particle size, polydispersity index and zeta potential were analyzed at 25 °C (Malven Zetasizer Nano-ZS analyzer; Malvern Instruments, Malvern, UK). Dynamic light scattering was used to determine the particle size (PZ) and polydispersity index (PDI) of nanosystems and polyplexes after filtering the samples with a 0.22μm cellulose acetate filter. The zeta potential (ZP) was determined by electrophoretic light scattering.

Morphological characterization was analyzed by transmission electron microscopy (TEM) using a Tecnai G2 Spirit BioTWIN transmission electron microscope (FEI Company, Eindhoven). Freshly prepared nanosystem solutions were placed on a formvar and carbon-coated carbon grid, and the grid was dried for five minutes. Samples were analyzed at 120 kV. Digital images were captured with a digital camera system (MegaView III–SIS).

### 2.7. Gel Retardation Assay

The ability of F68PEI to complex pDNA was examined by electrophoresis. Freshly prepared polyplexes (N/P 5–100) were applied on agarose gel (1% in tris acetate EDTA buffer) stained with GreenSafe Premium (NZYTech, PT). Electrophoresis was performed at a constant voltage (80 V) for 40 min. The gels were visualized using a StepOne^^®^^ detection system (Thermo Fisher Scientific, Winsford, UK).

### 2.8. Structural Stability of Polyplexes

The colloidal stability of polyplexes composed of F68PEI/F68 1:5 and pDNA (N/P 50) was determined in a physiological medium. Polyplexes were incubated in McCoy’s 5A or DMEM cell medium complemented with 10% (*v/v*) FBS. The change in the average hydrodynamic diameter was monitored after 0.5, 1, 2 and 4 h of incubation.

### 2.9. Cell Culture

The cytotoxicity and transfection potential of prepared polyplexes were evaluated in U2OS (human bone osteosarcoma epithelial) and SCC-9 (human oral squamous cell carcinoma) cells. The cells were incubated at 37 °C in a 5% CO_2_ atmosphere in appropriate cell culture medium (DMEM:F12 for SCC-9 and McCoy’s 5A medium, respectively) supplemented with 10% (*v/v*) heat-inactivated FBS, penicillin (U mL^−1^) and streptomycin (100 μg mL^–1^). Hydrocortisone (400 ng/mL) was added to DMEM:F12. Cells were grown in monolayers and detached by treatment with a 0.25% trypsin solution (Sigma, St. Louis, MO, USA).

### 2.10. Cell Viability

Cell viability was determined by Alamar Blue assay. The U2OS and SCC-9 cells were plated in 48-well plates at a density of 3.5 × 10^4^ and 3.0 × 10^4^ cells/well, respectively, and incubated for 24 h. Then, the medium was replaced, and the cells were exposed to the freshly prepared polyplexes (with 1 μg pCMV.Luc/well) for 4 h. The transfection medium was removed and replaced, and the cells were incubated for another 48 h. Cell media were removed, and cells were incubated at 37 °C with media containing 10% (*v/v*) resazurin sodium salt previously prepared at 1 mg/mL. After 1 h (SCC-9) or 4 h (U2OS) the absorbance of the cell medium was measured at 570 and 600 nm. The cell viability was calculated as described previously [17].

### 2.11. Transfection Activity

The transfection activity of different polyplex systems was determined by luminescence, using luciferase as a reporter gene (pCMV.Luc plasmid). The same transfection protocol was used as that described above for the cell viability assay. To determine luciferase gene expression, the cells were washed 2× with PBS solution, after which 100 μL of lysis buffer was added to each well. Luciferase expression was quantified by measuring luminescence based on the ATP cell viability assay based on luciferase. The total protein content of the lysates was determined by a DC protein assay. Results are expressed as relative light units per mg of total cell protein.

### 2.12. Statistical Analysis

A one-way ANOVA (α = 0.05) with Tukey multiple comparison tests was conducted using Graphpad PRISM 8.3.0 (GraphPad Software, San Diego, CA, USA) to assess the statistical significance of differences between groups (*p*-value < 0.05).

## 3. Results and Discussion

### 3.1. Copolymer Synthesis and Characterization

A multistep synthetic approach was used to conjugate branched PEI (bPEI1.8k) to Pluronic^®^ F68 (Figure 1a). In the Pluronic^®^ activation step, the reaction of F68 with excess 1,1‘-carbonyldiimidazole (CDI) resulted in partial activation of F68 hydroxy groups, as demonstrated by ^1^H NMR spectroscopy. The formation of F68CDI was confirmed by the appearance of typical imidazole multiplets in the aromatic region (*δ* 7.14, 7.67 and 8.4 ppm) and a multiple corresponding to PEO CH_2_ groups in the α position of the carboxyl group (*δ* 4.37 ppm) (Figure 1b). Integration of imidazole and PPO methyl peaks (*δ* 1.1 ppm) revealed that only approximately 30% of OH groups were converted into corresponding alkoxycarbonyl imidazole F68CDI. Structural changes in F68 in the intermediary and final synthetic products were also observed by IR spectroscopy. The appearance of an additional band at 1761 cm^−1^ typical of C=O stretching indicates the presence of a carbonyl group, as previously described for other CDI-activated Pluronics^®^ (Figure 1c) [18]. Subsequent nucleophilic substitution with bPEI1.8k yielded carbamate F68PEI. In the NMR spectrum, new broad multiplets appeared between 2.6 and 3.1 ppms, with observed peak shifts when compared with the spectrum of pure PEI (Figure 1b). A shift of the CH_2_ adjacent to the carbonyl group was also observed (*δ* 4.25 ppm). Analysis of the peak areas suggested the presence of two molecules of F68 per molecule of PEI. Considering the incomplete activation of F68 (approximately one of four OH groups), the stoichiometry of the F68PEI conjugate is 1:1 with the presence of an equal amount of unconjugated F68. Such Pluronic^®^-bPEI1.8k 1:1 conjugates have been observed in the case of another hydrophilic Pluronic^®^, F38 (HBL = 31), in which high cmc and the presence of non-aggregated activated Pluronic^®^ chains under the reaction conditions promote conjugation of the observed stoichiometry. On the contrary, the proximity of multiple activated PEO chains in micelles formed by Pluronics^®^ with lower cmc might promote the conjugation of multiple Pluronics^®^ with the same PEI molecule [10]. In the IR spectrum, new bands appeared at approximately 1600 and 3280 cm^−1^, corresponding to N-H vibrations (PEI). The ability of Pluronics^®^ to form nanomicelles stabilized by hydrophobic interactions within the core and the protective hydrophilic shell provided an interesting vehicle for delivery of therapeutic nucleic acids (NA) by grafting Pluronics^®^ with cationic polymers able to bind and condense genetic material such as pDNA. The possibility of extending these structural benefits of Pluronics^®^ to NA transfection has been extensively studied in the past two decades. A wide range of Pluronics^®^ with varying PPO/PEO ratios, molecular masses, and related physicochemical properties, such as HBL (hydrophilic/lipophilic balance) and cmc (critical micellar concentration), were conjugated with PEI characterized by varying degrees of branching or molecular mass. However, the use of different linkers, polymer feeding ratios and experimental conditions (solvent, temperature, polymer concentration and reaction time) often resulted in a variety of final cationic copolymers with varying degrees of polymerization, even when the same Pluronic^®^/PEI combination was used [9]. Whereas the hydrophilicity of F68 (HBL = 29) combined with the reaction conditions might provide some insight into the origin of the stoichemistry of F68PEI, in the case of F68 activated by bis-(trichloromethyl)-carbonate/N-hydroxysuccinimide, subsequent reaction with the large excess of bPEI1.8k led to the conjugation of several Pluronic^®^ molecules with one PEI [12].

Additionally, the amount of PEI in F68PEI was estimated by TNBS assay based on the reaction of primary amines [16]. The degree of PEI grafting determined by this method is in reasonable agreement with the composition of F68PEI observed by NMR.

### 3.2. Buffering Capacity of F68PEI

One of the crucial properties of PEI-based polyplexes is their ability to escape from endosomal system and deliver their therapeutic cargo into the cytoplasm. Although it is commonly accepted that they are taken up by cells via diverse endocytic mechanisms through interactions with adhesion acceptors, the mechanism of endosomal release is still debated [19]. The “proton sponge” hypothesis relies on the buffering capacity of unprotonated amino groups to absorb H^+^, causing increased Cl^-^ and water influx. Finally, resulting osmotic disbalance causes endosome burst and subequal polyplex release. As shown in Figure 2, F68PEI possesses almost twice the buffering capacity of the parent PEI, suggesting that the copolymer might have superior transfection capacity.

Although free PEI chains that are not engaged in pDNA complexation are essential for endosomal escape, a small proportion of lysosomes contain high amounts of PEI polyplexes, and lysosomal burst events are rare [20]. Apart from organelle osmotic swelling, PEI protonation can cause polymer swelling and result in transitory holes and pores, which facilitate membrane translocation. They arise as a result of the interactions of PEI with the lipid membranes of endolysosomal compartments caused by combined pressure of osmotic and polymer swelling [19]. This effect seems to be cell-dependent, and in some cases membrane–polymer interactions, can cause a leaky membrane, resulting in the accumulation of therapeutic NA within the organelles, leading to reduced transfection. A recent study suggested that PEI might influence polyplex intracellular trafficking and participate in the dislocation of therapeutic DNA across the nuclear membrane. The model system demonstrated that bPEI1.8k has the ability to stabilize nuclear membrane nanopores by interactions with the phospholipid bilayer and that it is accompanied by unpacking of polyplex, allowing for the nuclear entry of a therapeutic load [21]. On the other hand, the grafting of nanosystems with biocompatible hydrophilic polymer chains such as PEO is a common strategy to improve the longevity of nanovehicles under physiological conditions. Steric protection of such a hydrophilic envelope that protects nanosystems against premature sequestration can also prevent polyplex interactions with diverse phospholipid bilayers and decrease cell uptake. In the case of Pluronics^®^, this effect can be counterbalanced by the presence of a central hydrophobic block, which enables interactions with hydrophobic chains of the lipid bilayer [22]. The interactions of Pluronics^®^ with phospholipid bilayers strongly depend on their properties and concentrations. At concentrations below or close to the cmc, the polymer chains are free to insert themselves within the phospholipid chains and cause membrane thinning and fluidization, as was observed for hydrophilic F68 and F127 [22]. Although highly hydrophilic Pluronics^®^ such as F68 (HLB 29, 80% PEO) have protective effects on the lipid bilayer [23], depending on the used model and given sufficient time, F68 can disrupt and permeabilize the cell membrane [4]. Additionally, both free Pluronic^®^ chains and micelles can enter cells by caveolae- or clathrin-mediated endocytosis, respectively, as demonstrated for P85 [24]. Crucially, Pluronic^®^ interactions with cell membranes are also cell-type dependent [25]. Therefore, the F68 component of the copolymer or the addition of free F68 might contribute to cellular uptake and endosomal release by both specific and non-specific lipid membrane interactions. Considering that the cmc of Pluronic^®^-PEI systems with the addition of free Pluronic^®^ is within the same order of magnitude as the parent Pluronic^®^ [10], it is safe to conclude that the concentrations of F68PEI systems used in this study were below or close to the polymer cmc (4 mg/mL for F68), and the possible effects can therefore be attributed to free polymer chains or submicellar aggregates.

### 3.3. Polyplex Preparation and Characterization

Previous studies on the use of Pluronics^®^ in gene transfection revealed that they are able to improve the transfection of both naked pDNA and polycation pDNA polyplexes, predominantly by exploiting the stabilization provided by Pluronic^®^ micelles [26]. Furthermore, the addition of hydrophilic Pluronics^®^ (F68, F127) at higher concentrations (1–3%) markedly improved the stability and transfection of pDNA/PEI in the FBS-rich medium [27]. Similarly, the addition of free Pluronics^®^ to Pluronic^®^-PEI polyplexes improved stability and resulted in increased transfection [10]. However, these effects were mostly studied for the Pluronics^®^ with medium hydrophilicity and low cmc, such as P123 (HLB 8, cmc 0.025 % *w/w*) and P85 (HLB 16, cmc 0.03 *% w/w*). The addition of Pluronics^®^ with different HLBs (3–22) suggests that although improved transfection was observed for all studied combinations, the most considerable influence on pDNA transfection efficacy was exhibited by L35 (HLB 19) [28]. However, other studies have concluded that PEI conjugates of hydrophilic F68 still possess considerable transfection potential comparable to other less hydrophilic Pluronics^®^ [11,12]. Therefore, to further explore the potential of F68-based transfection agents, we decided to test the capacity of F68-bPEI1.8k conjugate as a pDNA delivery vehicle in the presence of free F68.

Polyplexes were prepared with F68PEI alone or in combination with free F68 (1:5 *w/w*) at increasing N/P ratios. The concentration of the total N atoms was estimated based on the amount of PEI determined by NMR and TNBS assays. After the addition of pDNA (1 μg), polyplexes were analyzed by electrophoresis (Figure 3).

The results of gel electrophoresis demonstrate that both polymer formulations are able to fully condense pDNA at N/P 10, indicating that the addition of free F68 did not influence the ability of F68PEI to stabilize pDNA as observed in previous studies [28].

### 3.4. Cytotoxicity of F68 Polyplexes

The cell viability of two cell lines, U2OS (human bone osteosarcoma epithelial) and SCC-9 (human oral squamous carcinoma), exposed to polyplexes prepared with F68PEI, F68PEI/F68 1:2 or F68PEI/F68 1:5 was determined by Alamar Blue test, a simple and non-toxic alternative to MTT, which was exclusively used in previous studies. Studies were performed with formulations prepared with N/P ratios between 5 and 100, and cell viability was compared to control bPEI25k- and the parent bPEI1.8k-based polyplexes (Figure 4).

The application of biocompatible nanovehicles is extremely important for successful therapeutic application, especially when systemic administration is concerned. All three formulations exhibit favorable biocompatibility at the polymer concentrations that correspond to low N/Ps, and compromised cell viability can be observed for N/P > 25. Such and effect of increasing concentrations of Pluronic^®^-PEI was previously described for different combinations of Pluronics^®^, PEIs and cells [29]. Up to N/P 25 cell viability is comparable with both PEI controls. This effect seems to be more pronounced in SCC-9 cells (Figure 4a). Apart from the association of increasing N/P with increased amounts of positive charge, cell viability was also influenced by the amount of added free F68. Because the formulations used in this study contain the same amount of F68PEI, the marked drop observed for F68PEI/F68 1:5 is related to the formulation that contains the highest amount of free F68. The influence of Pluronics^®^ on cell viability was previously observed but only when the PEI conjugate was based on Pluronics^®^ with low HLB [10,28]. Although biological activity has been extensively studied for a variety of Pluronics^®^, the polymers of the intermediate HLB are considered to possess the most potent activity relevant to anticancer therapy [4]. As previously discussed in Section 3.2, the optimal concentration range for biological activity is below or close to cmc, and a significant effect on cancer metabolism, mitochondrial and drug efflux transporter activity was observed, especially in multiple drug-resistant cancer cells [4,30,31]. Despite the modest biological activity of hydrophilic Pluronics^®^ such as F68 when compared with congeners with moderate HLB, the addition of F68 enhanced the ability of P86 micelles to inhibit the multidrug resistance protein MRP2-dependent drug efflux at the cmc of mixed P86/F68 micelles [32]. Whereas this might be relevant to the application of F68 in drug-based or combination therapies, the mechanism of the observed effects on cell viability caused by the formulation with the highest free F68 proportion is currently unknown. Additionally, studies using similar sub-cmc F68 concentrations, although conducted with different cell types, have not reported any significant effects on viability [33]. Other studies also demonstrated concentration- and cell-dependent responses to F68, and cytotoxic effects were mainly observed at concentrations above cmc [34]. These findings indicate that in this particular case, the observed effects might be the result of concerted activity of F68PEI and free F68.

### 3.5. Transfection Efficiency of F68 Polyplexes

The in vitro capacity of F68PEI-based formulations to transfect SCC-9 and U2OS cells was evaluated by complexing pCMV.Luc plasmid. Transfection experiments were conducted in cell media containing 10% heat-inactivated FBS.

The results based on luciferase expression reveal strong cell-dependent susceptibility to transfection by F68PEI-based polyplexes (Figure 5). The ability of F68PEI/DNA polyplexes to transfect SCC-9 cells is superior to that observed for both used controls (bPEI25k and bPEI1.8k), especially at N/P > 25. For F68PEI and F68PEI/F68 1:2, a clear dependence of transfection on increasing N/P ratios is observed. However, around N/P 50, the maximum transfection activity seems to be reached. A similar trend was observed for F68PEI/F68 1:5, with optimal transfection at N/P 75. In U2OS cells, transfection is comparable to that of PEI controls. For all the tested nanosystems, significant levels of transfection were observed only for N/P > 5. In the case of F68PEI and F68PEI/F68 1:2, a statistically significant increase in transfection capacity was observed for the highest tested N/P ratio (100). A more pronounced influence of increasing N/P on transfection activity was observed for polyplexes prepared with F68PEI/F68 1:5, and the most transfection was observed at N/P 75, as in the case of SCC-9 cells. The influence of the addition of free F68 on transfection is especially evident at higher N/Ps (50–100) for both cell lines, for which statistical relevance was observed in an intergroup comparison. Although reports on the influence of free F68 on polyplex transfection are scarce, pre-exposure of cells to F68 in sub-cmc concentrations enhanced the Turbofect transfection of HeLa cells [33].

### 3.6. Physicochemical Characterization of F68 Polyplexes

Size and zeta potential are essential parameters that govern the successful cellular internalization of polyplexes. Nanosystems smaller than 200 nm are preferentially taken up by cells, whereas positive charge facilitates interactions with the cell membrane and promotes internalization [5]. Therefore, we decided to study the influence of polyplex formulation on nanosystem size and zeta potential. We focused on detailed characterization of F68PEI and F68PEI/F68 1:5 based on the results obtained from the cell viability and transfection assays. Initial experiments conducted in ddH_2_0 demonstrate that pDNA complexation by both F68 systems results in reduction in nanosystem size, albeit with high polydispersity (Table 1). Under the given experimental conditions based on transfection with 1μg of pDNA, the concentration of polymers needed to achieve the desired N/P falls below the cmc value, presuming that the cmc of the parent Pluronic^®^ is not affected by conjugation with PEI or the addition of free Pluronic^®^ (as mentioned in the previous Sections). Therefore, the stabilization of the polyplex due to micelle formation can be ruled out in this case. A stabilizing effect was previously observed for Pluronic^®^-PEI systems based on polymers with low cmc [10,35]. In general, the use of F68 for the delivery of active pharmaceutical ingredients is uncommon, as it is plagued by high cmc when compared to other commonly studied Pluronics^®^, and reports describing its use in this capacity are rare. Nanomicelles with low cmc prevent the loss of therapeutic load upon systemic dilution and immune system clearance and maintain the integrity of therapeutic NAs by providing protection against nucleases [1,2]. Therefore, attempts have been made to improve its applicability by conjugation with hydrophobic moieties [36].

The addition of free Pluronics^®^ in such cases did not significantly change the size of the mixed polyplex micelles. In our study, the addition of free F68 (F68PEI/F68 1:5) reduced the average size of polyplexes to less than 200 nm, whereas at the corresponding N/P ratios, the polyplex formed by F68PEI remained above 200 nm.

Such size reduction could reflect the increase in total polymer concentration and increased aggregation processes as the concentration approaches cmc. However, no clear trends were observed in terms of polyplex size with increasing N/P. Changes were observed in the zeta potential of polymer solutions. Before adding pDNA, the expected increase in N/P ratio was observed, reflecting the increased concentration of positively-charged polymer. The addition of free F68 does not seem to considerably contribute to the additional screening of positive charge (see the values in brackets). pDNA complexation caused charge neutralization, and a decrease in zeta potential was observed until N/P 50, followed by a slight increase. Considering the higher transfection activity of F68PEI/F68 1:5, we prepared pDNA polyplexes with this polymer system in PBS in order to evaluate polyplex behavior under simulated physiological conditions. Similar polyplex sizes and changes in zeta potential were observed. Next, we focused on F68PEI/F68 1:5 at N/P 50, at which the highest transfection was observed without excessive loss of cell viability (80% in SCC-9 and 90% in U2OS). The size of polyplexes prepared in DMEM:F12 without the addition of inactivated FBS was approximately 130 nm. In complete cell media (DMEM:F12 and McCoy’s 5A), a slight increase in size was observed (around 210 nm) and remained stable for 4 h (time of cell exposure to polyplexes). A similar effect of complete medium was observed in bPEI25k and could be explained by the initial formation of protein corona [37]. Finally, the morphology of polyplexes was evaluated by TEM. Polyplexes were prepared at N/P 50 with F68PEI/F68 1:5 and compared to F68PEI polyplexes at the same N/P ratio. A variety of morphologies were formed by both polymer formulations (Figure 6).

Some of the morphologies, namely doughnut-like forms and thick rods, resemble those formed by PEO-grafted low-molecular-weight PEI [35], whereas thinner threads were identified as DNA coated with polymer, taking into account that the zeta potential is positive. Additionally, smaller (30–40 nm) and larger (70–80 nm) amorphous structures were detected. The observed morphologies are similar for both tested systems and reflect the polydispersity observed by DLS. Furthermore, even the formulation designated as F68PEI contains some free F68, as evidenced by NMR analysis. Currently, neither the nature and composition of these formations nor their role in the observed biological activity of F68-based polyplexes are known.

## 4. Conclusions

In this work, we studied the transfection of human bone osteosarcoma (U2OS) and oral squamous cell carcinoma (SCC-9) cells by copolymers composed of hydrophilic Pluronic^®^ F68 and low-molecular-weight PEI. The unique combination of a coupling agent and reaction conditions resulted in a new copolymer formulation that has not been tested as a gene delivery system to date. We demonstrated that the addition of free F68 results in polyplexes with improved physicochemical properties for a gene delivery system and increases transfection activity in a concentration-dependent manner. However, increased transfection at higher N/Ps (>50) is associated to decreased cell viability. Although this might not represent an undesired trait when considering applicability in antitumor treatment, it might pose problems when considering systemic administration. Therefore, these nanosystems can be considered for local administration. Furthermore, transfection activity strongly depends on the cell type and was greater in SCC-9 cells, where it highly surpassed the transfection activity of control bPEI25k. Similar studies have almost exclusively focused on Pluronic^®^-PEI polyplexes based on Pluronics^®^ with low and moderate HBL (<20), low cmc and/or with known anticancer activity. In this study, we demonstrated that hydrophilic Pluronic^®^ F68 can be used as a base for new pDNA delivery vehicles. Condensation of pDNA by the PEI component was, in this case, combined with the ability of the putative stabilizing influence of pure F68, as previously observed for other biologicals. Additionally, F68 component can contribute to improved transfection by transient cell membrane destabilization. We conclude that F68PEI-based polyplexes represent a potential base for further development of new cell-specific transfection agents.

## Figures and Tables

**Figure 1 polymers-14-05315-f001:**
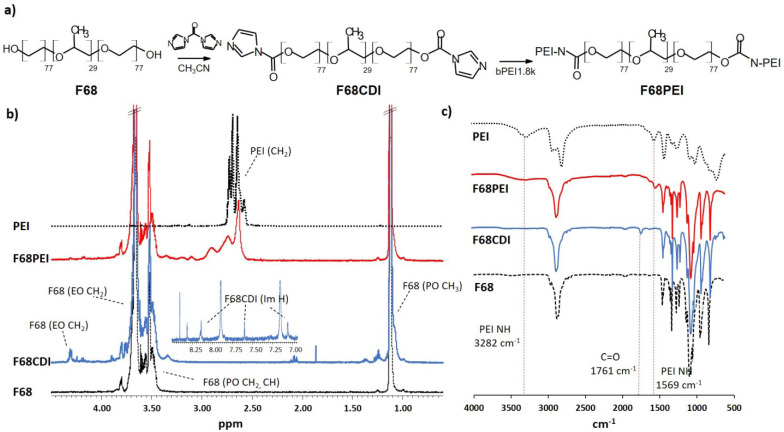
Synthesis and characterization of F68PEI copolymer. (**a**) Synthesis scheme; (**b**) ^1^H NMR spectra of F68, PEI, intermediary F68CDI and copolymer F68PEI. Insert shows the aromatic region with imidazole multiplets. Some multiplet resonances are assigned; (**c**) IR spectra of F68, PEI, intermediary F68CDI and copolymer F68PEI. Vertical lines indicate the appearance of characteristic peaks of new functional groups during F68PEI synthesis: introduction of a carboxylic group in F68CDI (1761 cm^−1^) and the appearance of vibrations characteristic of PEI in F68PEI (1569 and 3282 cm^−1^).

**Figure 2 polymers-14-05315-f002:**
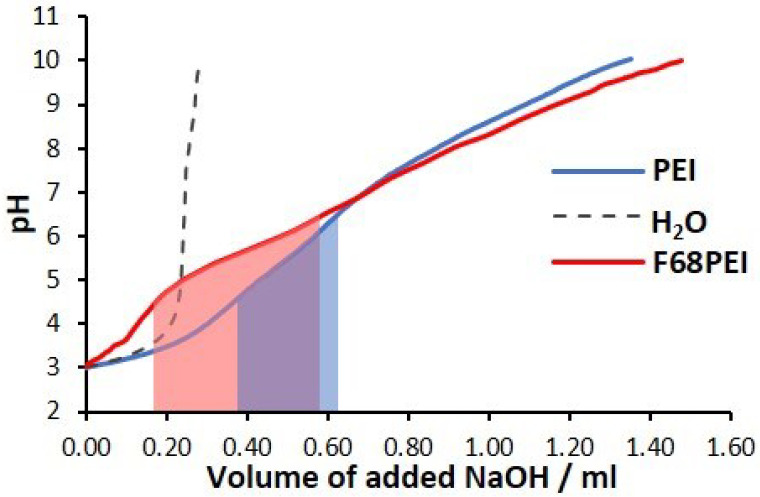
Buffering capacity of F68PEI and bPEI1.8k determined by titration with NaOH (0.1 M) within the range of pH 3–10. Colored inserts under the titration curves indicate NaOH volume needed to increase pH from 4.5 to 6.5, which is the physiological pH range of the endosomal system from lysosomes (pH 4.5) to early endosomal compartment (pH 6.5), in addition to indicating the buffering capacity, which is relevant for endosomal escape according to the “proton sponge” theory.

**Figure 3 polymers-14-05315-f003:**
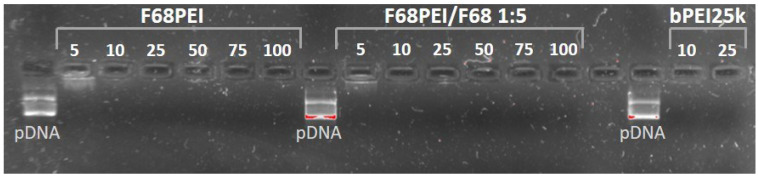
DNA complexation ability of F68PEI without and with the addition of free F68 (1:5 *w/w*) determined by agarose gel electrophoresis. For F68PEI and F68PEI/F68 1:5, N/P ratios between 5 and 100 were tested. For the control bPEI25k N/P ratios between 10 and 25 were tested.

**Figure 4 polymers-14-05315-f004:**
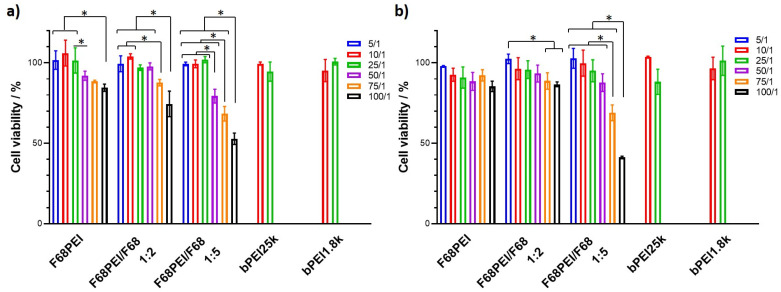
Cell viability of (**a**) SCC-9 and (**b**) U2OS cells exposed to polyplexes prepared with different F68PEI formulations: F68PEI, F68PEI/F68 1:2 and F68PEI/F68 1:5. Cells were exposed to the formulations for 4 h and analyzed after an additional 48 h. Data are expressed as the percentage of cell viability with respect to the control (nontreated cells) and shown as the mean ± standard deviation. * Statistically different groups (*p* < 0.05).

**Figure 5 polymers-14-05315-f005:**
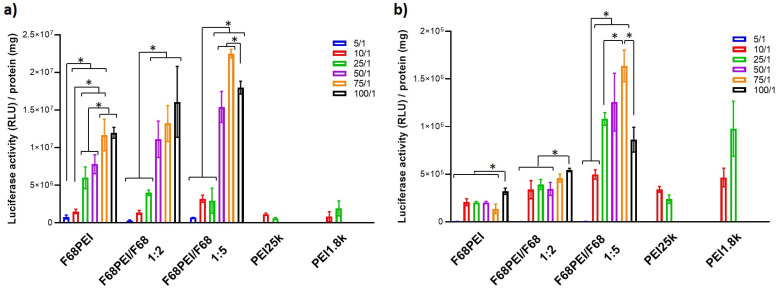
Transfection efficiency of F68PEI-based polyplexes in (**a**) SCC-9 and (**b**) U2OS cells. The following F68PEI formulations were tested: F68PEI, F68PEI/F68 1:2 and F68PEI/F68 1:5. Increasing polymer concentrations were mixed with 1 μg of pCMV.Luc plasmid to achieve the desired N/P ratios (5–100). Data are expressed as the relative light unit (RLU) of luciferase per mg of total cell protein and shown as the mean ± standard deviation. * Statistically different groups (*p* < 0.05).

**Figure 6 polymers-14-05315-f006:**
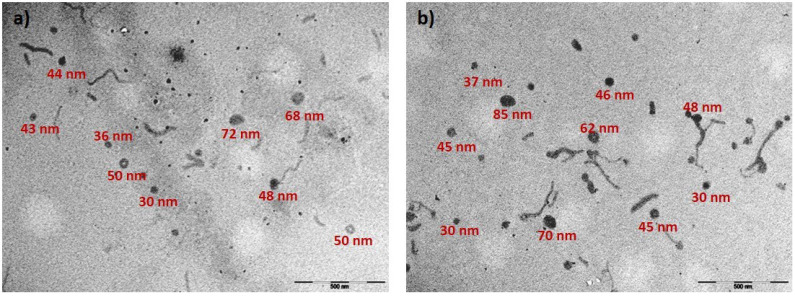
TEM images showing morphologies of polyplexes formed with (**a**) F68PEI and (**b**) F68PEI/F68 1:5 at N/P 50. The bar represents 500 nm.

**Table 1 polymers-14-05315-t001:** Characterization of the polyplexes formed by F68PEI with pDNA (ddH_2_O, pH = 7.3).

	N/P	Particle Size (nm)	PDI	Zeta Potential (mV)		N/P	Particle Size (nm)	PDI	Zeta Potential (mV)
**F68PEI**	5	210.5	0.41	51.4	**F68PEI/F68** **1:5**	5	151.4	0.57	43.3
	10	273.0 (591.3) *	0.52 (0.62)	41.3 (13.6)		10	188.6 (883.1)	0.50 (0.68)	40.4 (25.3)
	25	310.7 (486.1)	0.53 (0.52)	22.1 (35.3)		25	113.6 (582.6)	0.54 (0.68)	22.8 (31.6)
	50	277.0 (475.0)	0.44 (0.54)	41.3 (43.9)		50	183.4 (248.2)	0.39 (0.46)	20.1 (40.2)
	75	384.0	0.73	36.9		75	86.6	0.56	25.1
	100	217.4	0.49	40.4		100	101.1	0.55	28.7

* Values in brackets correspond to polymer solutions of the same concentration without pDNA addition.

## Data Availability

Not applicable.

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
