# Peer review of "Novel Non-Viral Vectors Based on Pluronic® F68PEI with Application in Oncology Field"

_polymers, 2022, doi:10.3390/polym14235315_

Round 1
Reviewer 1 Report
In this manuscript, the authors prepared new copolymer by conjugation F68 to branched PEI1.8k and studied their transfection capacity under sub-cmc conditions. Furthermore, the influence of addition of free F68 to F68PEI was investigated. However, there are several issues that do not allow me to recommend this manuscript for publication.
1. In figure 2, the line type of H2O and F68PEI are similar.
2. In figure 1 a), it should be bPEI1.8k, not bPEI2k, no bPEI2k appeared in this manuscript.
3. In figure 4 and figure 5, the statistically difference marked in the figures are not clearly.
4. The author can add confocal data to observe the intercellular distribution of F68PEI polyplexes.
5. The author need to discuss more about the advantages of F68 compared with other Pluronics.
6. What’s the highlight of this manuscript?
Author Response
Manuscript ID: polymers-2039160
Title: Novel non-viral vectors based on PluronicF68:PEI with application on the oncology field
Dear Editor,
Doctor Lionel Lin
The authors appreciate the careful review and greatly acknowledge the comments that the Reviewer has provided on our manuscript. We have carefully revised the manuscript and have made the recommended changes and answered in detail to the questions raised. Moreover, to help us reviewing the manuscript we have included Marília Dourado. Additional information was also added when appropriate. All changes made to the text are highlighted in green colour and marked up using the “Track Changes” function.
# Reviewer 1
In this manuscript, the authors prepared new copolymer by conjugation F68 to branched PEI1.8k and studied their transfection capacity under sub-cmc conditions. Furthermore, the influence of addition of free F68 to F68PEI was investigated. However, there are several issues that do not allow me to recommend this manuscript for publication.
- In figure 2, the line type of H2O and F68PEI are similar.
Response: Thank you for the observation. We have changed the color of lines to make them more distinguishable.
- In figure 1 a), it should be bPEI1.8k, not bPEI2k, no bPEI2k appeared in this manuscript.
Response: Thank you for the remark. We have corrected this in Figure 1.
- In figure 4 and figure 5, the statistically difference marked in the figures are not clearly.
Response: Thank you for the observation. We have presented statistically different groups in a more visible manner.
- The author can add confocal data to observe the intercellular distribution of F68PEI polyplexes.
Response: We agree that confocal imaging would add another level of information to our understanding of this class of polyplexes. Unfortunately, currently we are not in a position to conduct these experiments. However, considering that the luciferase luminescence assay used to determine transfection activity is performed in cell lysates after thorough washing with PBS, we are confident that the formulation was internalized and efficiently delivered the nucleic acid to the target site (nucleus).
- The author need to discuss more about the advantages of F68 compared with other Pluronics.
Response: We added a short discussion of F68 use and advantages in the Introduction section (Page 2).
Added section reads:
Although biological effects of F68 in cancer cells were not investigated in greater details, due to its hydrophilicity and biocompatibility it is widely used as surfactant, and solubilizing and emulsifying agent in pharmaceutical and cosmetic industries. It can improve oral availability of hydrophobic drugs [13] and stabilize liquid formulations containing biologicals like antibodies and proteins although the exact mechanism of observed stabilizing activity remains unknown [14]. Recently, several antibody formulations containing F68 were approved for clinical application [15].
- What’s the highlight of this manuscript?
Response: In order to highlight the results of our study we introduced a following text to the Conclusion section (Page 13):
So far, similar studies were almost exclusively focused on Pluronic-PEI polyplexes based on Pluronics with low and moderate HBL (<20), low cmc and/or with known anticancer activity. In this study we have demonstrated that hydrophilic Pluronic F68 can be used as a base for new pDNA delivery vehicles. Condensation of pDNA by the PEI component was in this case combined with the ability of the putative stabilizing influence of pure F68 as previously observed for other biologicals. Additionally, F68 component could contribute to improved transfection by transient cell membrane destabilization.

Reviewer 2 Report
In this study, Silva et al. reported the combination of PluronicF68 with PEI at different ratios to fine-tune the polyplexes for optimizing the transfection activity. Although this work is doing ok, the manuscript is not quite well written and needs further improvement:
1. Figure 2 is confusing. I hardly recognize the corresponding curve to each group. Better use different color to represent them
2. Session 3.2 and 3.4 contain too much discussion that makes readers hard to follow the results
3. “Slight movement of pDNA 344 at N/P 5 suggests that the positive charge might be partially screened by hydrophilic segments 345 of F68 and, therefore, not sufficient to fully complex/condensate pDNA”. The movement seems not significant and some bands are showing saturated intensity. It is difficult to draw this conclusion
4. The significant difference (i.e., a,b,c,d,e) symbols are not defined and they seem to be randomly distributed above the graph. I cannot identify the difference clearly.
5. Texts from all graphs are too small
6. Figure 6 should be described as TEM images.
7. The particles from table 1 seem very heterogeneous in terms of PDI. It is hard to convince readers that the polyplexes are well prepared. The cellular entry of particles also depends on the size effect.
Author Response
Manuscript ID: polymers-2039160
Title: Novel non-viral vectors based on PluronicF68:PEI with application on the oncology field
Dear Editor,
Doctor Lionel Lin
The authors appreciate the careful review and greatly acknowledge the comments that the Reviewer has provided on our manuscript. We have carefully revised the manuscript and have made the recommended changes and answered in detail to the questions raised. Moreover, to help us reviewing the manuscript we have included Marília Dourado. Additional information was also added when appropriate. All changes made to the text are highlighted in green colour and marked up using the “Track Changes” function.
# Reviewer 2
In this study, Silva et al. reported the combination of PluronicF68 with PEI at different ratios to fine-tune the polyplexes for optimizing the transfection activity. Although this work is doing ok, the manuscript is not quite well written and needs further improvement:
- Figure 2 is confusing. I hardly recognize the corresponding curve to each group. Better use different color to represent them
Response: Thank you for the remark. We have used different colors to mark individual groups.
- Session 3.2 and 3.4 contain too much discussion that makes readers hard to follow the results
Response: We thank the reviewer for the comment. In this paper we have decided to present results and discussion in the same Section, which might have made it more complex to follow the results presentation. However, after careful consideration, we have concluded that we needed to include such an elaborate discussion in order to properly interpret and justify our results.
3.“Slight movement of pDNA 344 at N/P 5 suggests that the positive charge might be partially screened by hydrophilic segments 345 of F68 and, therefore, not sufficient to fully complex/condensate pDNA”. The movement seems not significant and some bands are showing saturated intensity. It is difficult to draw this conclusion
Response: We agree with the reviewer and have removed the sentence in the manuscript.
- The significant difference (i.e., a,b,c,d,e) symbols are not defined and they seem to be randomly distributed above the graph. I cannot identify the difference clearly.
Response: Thank you for the observation. We have presented statistically different groups in a more visible manner.
- Texts from all graphs are too small
Response: Thank you for the remark. We have increased letter fonts in the figures.
- Figure 6 should be described as TEM images.
Response: Thank you for the suggestion. We have corrected the legend.
It now reads:
TEM images showing morphologies of polyplexes formed with a) F68PEI and b) F68PEI/F68 1:5 at N/P 50. The bar represents 500 nm.
- The particles from table 1 seem very heterogeneous in terms of PDI. It is hard to convince readers that the polyplexes are well prepared. The cellular entry of particles also depends on the size effect.
Response: Thank you for the comment. We share your concern about the formulation polydispersity. In order to test the influence of polyplex formation protocol, we also prepared polyplexes by addition of polymer to pDNA solution (reverse order of the one we described in the paper). However, the polyplex size and polydispersity were very similar. We would also like to stress the fact that the formulations were not filtered before cell exposure. In our unpublished experiments we have tested the influence of polymer concentrations on the size and polydispersity, and have obtained monodispersed micelles only for concentrations well above those used in our transfection experiments. Our research team has previously described aqueous formulations of pDNA polyplexes with high PDI (>0.3) that exhibited good in vitro transfection activity (https://doi.org/10.1016/j.actbio.2016.10.015).
Moreover, in complete cell culture medium, the size was around 200 nm, which is deemed acceptable for successful cellular uptake. Additionally, the polyplexes with the protein corona formed in the cell medium with the addition of FBS present the effective formulations with which the cells interact, and mimic relevant biological matrices.

Reviewer 3 Report
The author have done an interesting work on designing non-viral vectors using PluronicF68:PEI for its biomedical use in cancer. The article indeed is of a great interest and the experiments have been performed thoroughly. However there are certain minor comments the authors should address.
1. In figure 1, the authors have nicely presented the schematic diagram of the reaction. I would suggest that the authors also label the carboxylic group at 1761 cm-1 and for PEI at (1569 and 3282 cm-1) in the figure itself similar as done in the NMR.
2. In the cell survivability assay why were the SCC9 cells incubated with Alamar Blue for 1h whereas U2OS for 4h? In figure 4 , it would be good to mention the time of treatment in figure legend i.e. 4h or 48h?
3. Pluronic have recently gained a lot of interest in designing nanovehicles for application in oncology and the authors must highlight them such as articles “ Hyaluronic-Acid-Tagged Cubosomes Deliver Cytotoxics Specifically to CD44-Positive Cancer Cells” and article “Affimer Tagged Cubosomes: Targeting of Carcinoembryonic Antigen Expressing Colorectal Cancer Cells Using In Vitro and In Vivo Models”.
4. The authors should improvise the figure 6 legend and mention that these images are from TEM images for ease of a reader to understand. Eg. TEM images showing morphologies of polyplexes formed with a) F68PEI and b) F68PEI/F68 1:5 at N/P 50. The bar represents 500 nm.
Author Response
Manuscript ID: polymers-2039160
Title: Novel non-viral vectors based on PluronicF68:PEI with application on the oncology field
Dear Editor,
Doctor Lionel Lin
The authors appreciate the careful review and greatly acknowledge the comments that the Reviewer has provided on our manuscript. We have carefully revised the manuscript and have made the recommended changes and answered in detail to the questions raised. Moreover, to help us reviewing the manuscript we have included Marília Dourado. Additional information was also added when appropriate. All changes made to the text are highlighted in green colour and marked up using the “Track Changes” function.
# Reviewer 3
The author have done an interesting work on designing non-viral vectors using PluronicF68:PEI for its biomedical use in cancer. The article indeed is of a great interest and the experiments have been performed thoroughly. However there are certain minor comments the authors should address.
- In figure 1, the authors have nicely presented the schematic diagram of the reaction. I would suggest that the authors also label the carboxylic group at 1761 cm-1 and for PEI at (1569 and 3282 cm-1) in the figure itself similar as done in the NMR.
Response: Thank you for the remark. We have introduced assignation for new IR peaks into the figure.
- In the cell survivability assay why were the SCC9 cells incubated with Alamar Blue for 1h whereas U2OS for 4h? In figure 4 , it would be good to mention the time of treatment in figure legend i.e. 4h or 48h?
Response: Alamar Blue is based on reduction of resazurin to resorufin under the reductive intracellular conditions, and cell health is proportional to the produced fluorescence. It does not interfere with the electron transport chain. We have used the test according to the producer instructions which advised exposure of 1 to 4 h. However, the reduction rate of resazurin depends on cellular metabolic rate and development of resorufin color varies from cell to cell (https://doi.org/10.1046/j.1432-1327.2000.01606.x, DOI: 10.1016/s0022-1759(97)00043-4) and exposure time to resazurin should be optimized for each cell line. In our case, it took 1 h for SCC9 and 4h for U2OS to change the color.
We have introduced the exposure time in the legend of Figure 4. It now reads:
Cell viability of a) SCC9 and b) U2OS cells exposed to polyplexes prepared with different F68PEI formulations: F68PEI, F68PEI/F68 1:2 and F68PEI/F68 1:5. Cells were exposed to the formulations for 4 h, and analyzed after additional 48h. Data are expressed as the percentage of cell viability with respect to the control (nontreated cells), shown as the mean ± standard deviation. Statistically different groups (p<0.05) are denoted by the letter code: a N/P 5; b N/P 10; c N/P 25; d N/P 50; e N/P 75; f N/P 100.
- Pluronic have recently gained a lot of interest in designing nanovehicles for application in oncology and the authors must highlight them such as articles “ Hyaluronic-Acid-Tagged Cubosomes Deliver Cytotoxics Specifically to CD44-Positive Cancer Cells” and article “Affimer Tagged Cubosomes: Targeting of Carcinoembryonic Antigen Expressing Colorectal Cancer Cells Using In Vitro and In Vivo Models”.
Response: Thank you for the suggestion. We have added relevant references in Introduction section. It now reads:
Different combinations of these two components result in a wide range of physicochemical and biological properties which can be explored for improved drug and gene delivery [2, 3].
Ref 3: DOI: 10.1021/acs.molpharmaceut.2c00439
- The authors should improvise the figure 6 legend and mention that these images are from TEM images for ease of a reader to understand. Eg. TEM images showing morphologies of polyplexes formed with a) F68PEI and b) F68PEI/F68 1:5 at N/P 50. The bar represents 500 nm.
Response: Thank you for the suggestion. We have corrected the legend as proposed.

Round 2
Reviewer 1 Report
Accept
Reviewer 2 Report
The authors have addressed my comments appropriately.